# Comparative Proteomic Analyses of Susceptible and Resistant Maize Inbred Lines at the Stage of Enations Forming following Infection by Rice Black-Streaked Dwarf Virus

**DOI:** 10.3390/v14122604

**Published:** 2022-11-23

**Authors:** Rong Wang, Kaitong Du, Tong Jiang, Dianping Di, Zaifeng Fan, Tao Zhou

**Affiliations:** 1Institute of Medicinal Plant Development, Chinese Academy of Medical Sciences and Peking Union Medical College, Beijing 100193, China; 2State Key Laboratory for Agro-Biotechnology and Department of Plant Pathology, China Agricultural University, Beijing 100193, China; 3Institute of Plant Protection, Hebei Academy of Agriculture and Forestry Science, Baoding 071000, China

**Keywords:** rice black-streaked dwarf virus, *Zea mays*, comparative proteome, enation structure, differentially expressed protein, stress

## Abstract

Rice black-streaked dwarf virus (RBSDV) is the main pathogen causing maize rough dwarf disease (MRDD) in China. Typical enation symptoms along the abaxial leaf veins prevail in RBSDV-infected maize inbred line B73 (susceptible to RBSDV), but not in X178 (resistant to RBSDV). Observation of the microstructures of epidermal cells and cross section of enations from RBSDV-infected maize leaves found that the increase of epidermal cell and phloem cell numbers is associated with enation formation. To identify proteins associated with enation formation and candidate proteins against RBSDV infection, comparative proteomics between B73 and X178 plants were conducted using isobaric tags for relative and absolute quantitation (iTRAQ) with leaf samples at the enation forming stage. The proteomics data showed that 260 and 316 differentially expressed proteins (DEPs) were identified in B73 and X178, respectively. We found that the majority of DEPs are located in the chloroplast and cytoplasm. Moreover, RBSDV infection resulted in dramatic changes of DEPs enriched by the metabolic process, response to stress and the biosynthetic process. Strikingly, a cell number regulator 10 was significantly down-regulated in RBSDV-infected B73 plants. Altogether, these data will provide value information for future studies to analyze molecular events during both enation formation and resistance mechanism to RBSDV infection.

## 1. Introduction

Maize (*Zea mays*) is one of the most important crops worldwide. It can be utilized not only for feeding humans and livestock, but also for producing alcohol, sucrose, starch and industrial raw materials [1]. Maize rough dwarf disease (MRDD) is a serious viral disease that occurs around the world, especially in the summer maize-growing regions of China, resulting in heavy yield losses of up to 100% [2,3]. To date, four fiji-viruses, maize rough dwarf virus (MRDV) [4], Mal de Rio Cuarto virus (MRCV) [5], rice black-streaked dwarf virus (RBSDV) [6] and southern rice black-streaked dwarf virus (SRBSDV) [7], have been reported to cause MRDD. In China, RBSDV is the most prevalent causal agent of MRDD [6,8]. Maize plants infected by RBSDV show stunting, vein swellings, dark green coloring of leaves and suppression of flowering. Typical enation (or tumor) symptoms also occur on the back of the leaf and sheath of susceptible maize lines [2,9]. In addition, RBSDV can infect rice (*Oryza sativa*), wheat (*Triticum sativum*), oat (*Avena sativa*), barley (*Hordeum vulgare*) and some gramineous weeds, which can also cause losses in production [10,11]. RBSDV is naturally transmitted by the small brown planthopper (*Laodelphax striatellus*) in a persistent propagative manner [12]. Current management strategies for controlling RBSDV mainly include delaying sowing date, reducing population of the small brown planthopper and growing virus-resistant cultivars. Several genetic loci tightly linked to the MRDD resistance, such as two RBSDV resistance quantitative trait locus (QTLs), qMRD8 and qMrdd1 and two simple sequence repeats (SSR), 6F29R29 and 6F34R34, have been reported [13,14,15,16]. Most recently, *Rab GDP dissociation inhibitor alpha* (*ZmGDIα*) was identified as a quantitative recessive resistance gene to RBSDV [17]. However, maize genes and proteins involved in enation formation caused by RBSDV still remain largely unknown.

Considering the importance of RBSDV in maize, several genome-wide studies on maize gene and protein expressions have been conducted to analyze how susceptible maize plants respond to RBSDV infection [18,19,20]. A microarray study showed that various stress response-related genes, such as *PR1*, glutathione-S-transferase, *MYB* transcription factor family, *WRKY* family, development-related genes and auxin response genes, were altered dramatically during the preliminary stage of RBSDV infection in maize [18]. Comparative proteomics analyses identified a large number of DEPs involved in glycolysis, starch metabolism and plant defense response at the late stage of RBSDV infection, such as G-proteins, antioxidant enzymes, lipoxygenases and UDP-glucosyltransferase BX9 [19]. Meanwhile, proteomics analyses of early stage of RBSDV infection showed that most of altered proteins were strongly associated with cyano-amino acid metabolism, protein processing in ER and ribosome-related pathways [20]. However, few studies have been performed to compare the differences between susceptible and resistant maize cultivars in response to RBSDV. Following RBSDV infection, the susceptible maize inbred line B73 plants display typical symptoms including stunting of plants, dark green leaf color, enation on the back of the leaf and sheath and so on, while the resistant maize inbred line X178 plants rarely show symptoms. Enation formation along leaf veins is a typical early disease symptom of RBSDV infection in susceptible maize lines and cultivars. Thus, identification and characterization of regulated genes or proteins by RBSDV infection at early stage of symptoms formation in susceptible and resistant maize lines would help analyze the mechanism of symptoms formation and identify essential factors for host resistance.

Here, we firstly uncovered the microstructures of enations in leaves of RBSDV-infected maize plants. Then, comparative proteomics analyses between RBSDV-susceptible line B73 and resistant line X178 at the stage of enation forming were conducted through isobaric tags for relative and absolute quantitation (iTRAQ), followed by mass spectrometry analyses [21,22]. The results suggest that the defense or stress-related, development-related and photosynthesis-related DEPs were profoundly affected by RBSDV infection.

## 2. Materials and Methods

### 2.1. Plant Materials and RBSDV Inoculation

RBSDV-infected wheat plants were collected as viral source from Hebei province, China. Seeds of maize inbred lines B73 and X178 were sown in pots filled with commercial nutrient soil (Pindstrup Mosebrug A/S, Denmark) and grown in a greenhouse at 25 °C under natural light. When the seedlings of B73 and X178 were at three-leaf stage, they were exposed to *Laodelphax striatellus* carrying RBSDV (seven insects on each seedling) for 3 days in inoculation chambers. Seedlings inoculated with virus-free *L. striatellus* were mock control. The inoculated plants were transferred into soil in the greenhouse after completely removing the insects and grown in a glasshouse at 25 °C under natural sunlight.

### 2.2. Reverse Transcription (RT)-PCR Detection of RBSDV and Quantitative Analysis of RBSDV by Quantitative RT-PCR (RT-qPCR)

Total RNA was extracted from the maturing zones [23] of the first upper newly developed leaves which were harvested at 15 days post inoculation (dpi) with TRNzol reagent (Tiangen, Beijing, China) and then treated with RNase-free DNase I (TaKaRa, Dalian, China). First strand cDNA was synthesized using 1 μg total RNA, M-MLV reverse transcriptase (Promega, Madison, WI, USA) and random hexamer primer according to the manufacturer’s instructions. PCR was performed using S7-specific primers (qRP-F: 5′-GCTCCTACTGAGTTGCCTGTC-3′ and P1-R: 5′-TCAGCAAAAGGTAAAGGAAGG-3′) [24] and *Ex* Taq (TaKaRa, Dalian, China) according to the manufacturer’s instructions. RT-qPCR was performed using ten-fold diluted cDNA product, primers qRP-F and P1-R and FastSYBR mixture (CWBIO, Beijing, China) with ABI PRISM 7500 sequence detection system (Applied Biosystems Inc., Foster City, CA, USA). Viral concentrations were measured as described previously [19,24] via comparing the threshold cycles (Ct) value of each sample with the standard curve, constructed based on Ct values of serially diluted recombinant plasmid containing a partial fragment of RBSDV S7.

### 2.3. Scanning Electron Microscopy (SEM) and Paraffin Sections

Fresh leaf tissues of around 3 mm × 4 mm in size were harvested from the maturing zones of the first upper newly developed leaves. For SEM, the tissues were fixed in a mixture of 2.5% glutaraldehyde and 0.2 M sodium phosphate buffer (pH7.4). The fixed tissues were dehydrated through a standard ethanol series, then at the critical point dried with CO_2_ and coated with gold in 108 auto sputter coater (Cressington Scientific Instruments Ltd, Watford, UK). All specimens were examined and photographed using a scanning electron microscope at an accelerating voltage of 15 kV (Hitachi S-3400N, Tokyo, Japan). For paraffin sections, the tissues were fixed, dehydrated, processed and paraffin embedded as described previously [25]. Cross sections of tissues with 10 μm thickness were cut on a rotary microtome (Leica RM2235, Wetzla, Germany) and transferred on slides for observation under microscope (Nikon 80i, Tokyo, Japan).

### 2.4. Protein Preparation for iTRAQ Experiments

Total protein was extracted from the maturing zones [23] of the first upper newly developed leaves, each pool containing three pieces of leaves deriving from three plants, following the method previously described [26]. Briefly, leaf tissues were ground in liquid nitrogen and treated with lysis buffer (2 M thiourea, 7 M urea, 4% 3-((3-cholamidopropyl) dimethyl-ammonio)-1-propanesulfomate, pH8.5) followed by 1 min ultrasonication (0.2 s/2 s, on/off, amplitude 22%). Then protein was extracted by trichloroacetic acid/acetone method, as described by Sheoran et al. [27]. Protein concentration in each sample was measured through Bradford assay (Bio-Rad) as described by Kruger [28]. Protein samples were digested into peptides by trypsin and labeled with iTRAQ reagent eight-plex kit according to the manufacturer’s instructions (Applied Biosystems Inc., Foster City, CA, USA). The label number for each sample was as follows: 114 for the sample of RBSDV-infected X178, 115 for the sample of mock-inoculated X178, 116 for the sample of RBSDV-infected B73 and 117 for the sample of mock-inoculated B73.

An equal number of labeled samples were pooled and separated using a reverse chromatography column on a Rigol L-3000 system (Rigol, Beijing, China), as described by Chen et al. [26]. Briefly, the pooled peptides were dissolved in 50 μL mobile phase A (2% (*v*/*v*) acetonitrile in ddH_2_O, pH10) and then centrifuged at 14,000× *g* for 20 min. The supernatant was loaded into the column and then eluted stepwise by injecting mobile phase B (98% (*v*/*v*) acetonitrile in ddH_2_O, pH10). The flow rate was set at 700 µL min^−1^ and fractions were collected at 1.5 min intervals. The collected fractions were combined to produce 10 fractions and lyophilized.

### 2.5. Triple Quadrupole Time-of-Flight Tandem Mass Spectrometer

The fractionated peptides were analyzed through a triple quadrupole TOF 5600 System (AB Sciex, Concord, ON, USA) following the protocol described previously [29]. Data were acquired with an ion spray voltage of 2.3 kV, the temperature of an interface heater at 150 °C and curtain gas and nebulizer gas at 206.85 and 41.37 Kpa, respectively. The MS was generated by TOF-MS scans operating with a RP of 30,000_fwhm_. Information-dependent acquisition was performed as previously described [29]. For collision-induced dissociation, all precursor ions were performed at a Rolling collision energy setting. Dynamic exclusion was set at half of peak width (*c.* 8 s) and the precursor was then refreshed off the exclusion list [29].

### 2.6. Database Searching and Protein Quantitation

Raw data produced by MS were analyzed through Paragon and ProGroup algorithms in PROTEINPILOT software 4.0 (AB Sciex, Foster City, CA, USA) according to the previously published method [30]. Parameters was set as previously described [26]. Maize protein database was downloaded from NCBI (www.ncbi.nlm.nih.gov). The integrated tools in PROTEINPILOT software were used to control false discovery rate (FDR) under 1% and the ProGroup algorithm was used to identify peptides form tandem MS data.

Protein quantification was performed with PROTEINPILOT software as previously described [30] to estimate the relative abundance of iTRAQ-labeled peptides as well as the corresponding proteins. Corrections were made using impurity of iTRAQ reagents provided by the manufacturer. For further corrections, iTRAQ ratios were normalized by the bias correction factors.

### 2.7. Functional Analyses and Subcellular Localization

The identified proteins were classified based on their molecular and biological functions predicted by gene ontology (GO) annotations. GO analysis was carried out using BLAST2GO software [31]. Subcellular localization of identified proteins was predicted according to the information available in the WolfPsort and UniProt database [32,33].

## 3. Results

### 3.1. RBSDV Infections in Two Maize Inbred Lines

Seedlings of a susceptible maize inbred line B73 and a resistant line X178 were individually inoculated with RBSDV using *L. striatellus* carrying RBSDV. By 9 dpi, white streak started to appear in the upper leaves of RBSDV-infected B73 seedlings, but not in leaves of RBSDV-infected X178 plants. By 15 dpi, the white streaks in B73 leaves changed to enations, while no obvious symptoms were observed in leaves of RBSDV-infected X178 (Figure 1a). Viral infection was determined by RT-PCR at 15 dpi (Appendix A). The accumulation levels of viral RNA in RBSDV-infected B73 and X178 at 15 dpi were also determined using RT-qPCR, showing that there were around 10^4^ times more viral RNA per 100 ng total RNA of RBSDV-infected B73 leaves than in X178 (Figure 1b).

### 3.2. Cell Number Increasing Associates with Enations Formation in RBSDV-Infected B73 Leaves

According to previous studies, enations on leaves induced by RBSDV derived from hypertrophy of phloem cells, through expansion and multiplication of cells [34]. To further investigate the formation pattern of enations in RBSDV-infected maize leaves, surface morphological structure of leaves in RBSDV-infected or mock-inoculated B73 and X178 were observed. The photomicrographs showed that epidermal cells along leaf veins in leaf abaxial surface became smaller and irregular and cell numbers were significantly increased in RBSDV-infected B73 compared to mock-inoculated B73, while these changes were not observed on RBSDV- or mock-inoculated X178 (Figure 2a,b). These results showed that both the cell arrangement and cell number of epidermal cells in enations were altered by RBSDV infection in susceptible B73 plants, but not in resistant X178 plants. Epidermal cells in leaf adaxial surface did not show any difference between RBSDV- or mock-inoculated B73 and X178 (Appendix A), which were consistent with previous descriptions of enations formed on the abaxial surface, but not the adaxial surface of maize leaves following RBSDV infection [34].

To obtain the inside structure information of enations, cross sections of leaves were observed through paraffin sections via microscope. The micrographs showed that the structure of “Kranz anatomy” was destroyed in RBSDV-infected B73, but not in RBSDV-infected X178 (Figure 2c). The bundle sheath cells were arranged irregularly and most space of xylem cells was occupied by significantly increased numbers of phloem cells in RBSDV-infected B73, while the structure of “Kranz anatomy” in leaves of mock-inoculated B73, RBSDV-infected X178 and mock-inoculated X178 remains as normal (Figure 2c). Based on these results, it is suggested that the enations in RBSDV-infected B73 leaves may be caused by the excessive proliferation of epidermal cells and phloem cells.

### 3.3. Identification of DEPs Responsive to RBSDV Infection in B73 and X178

To identify proteins that might be associated with the enation formation in B73 leaves, comparative proteomic analyses were performed at the early stage of enation forming on B73 and X178 plants. The maturing zone of maize leaves (Figure 3a), in which transcriptional patterns of genes showed little variation [23], were used for proteomics analysis, ensuring that most of the identified DEPs are in response to RBSDV infection but not caused by leaf development. Results of iTRAQ analyses revealed that 260 and 316 proteins were significantly changed (Fold change ratio >1.5 or <0.67, *p* values < 0.05) in B73 and X178 plants, respectively, following RBSDV infection (Appendix A). In B73, 116 DEPs (*c.* 45%) were up-regulated and 144 DEPs (*c.* 55%) were down-regulated. In RBSDV-infected X178, 164 DEPs (*c.* 52%) were up-regulated and 152 DEPs (*c.* 48%) were down-regulated (Figure 3b). Among the identified DEPs, 105 DEPs were identified in both RBSDV-infected B73 and X178, of which 32 DEPs were up-regulated and 50 DEPs were down-regulated, suggesting that common responsive pathways were induced by RBSDV in both B73 and X178. The other 23 DEPs showed opposite up- or down-regulated in B73 and X178 by RBSDV infection (Figure 3c, Appendix A).

### 3.4. Gene Function Annotation and Protein Subcellular Localization Prediction

The identified DEPs were classified according to their GO annotations or their known biological functions. These DEPs could be functionally classified into 13 groups: biosynthetic process, development, energy metabolism, metabolic process, oxidation reduction, phosphorus metabolic process, photosynthesis, protein folding, proteolysis, response to stress, signal transduction, translation, transport and unknown (Figure 4). In B73, 71 DEPs were involved in metabolic process, accounting for 27.31% and 43 and 41 DEPs were involved in response to stress and biosynthetic process, accounting for 16.54% and 15.77%, respectively (Figure 4a). In X178, 92 DEPs were involved in metabolic process, accounting for 29.11% and 66 and 53 DEPs were involved in response to stress and biosynthetic process, accounting for 20.89% and 16.77%, respectively (Figure 4b). Detailed functional category annotations of the up- and down-regulated proteins are shown in Figure 4c,d. Intriguingly, though similar functional classification patterns of most DEPs were found in both susceptible B73 and resistant X178 in response to RBSDV infection, DEPs of oxidation reduction were totally up-regulated in B73 while DEPs of energy metabolism were all down-regulated in X178 following RBSDV infection (Figure 4c,d).

Subcellular localization of DEPs was predicted using the GO and UniProt database. The identified DEPs localized in various compartments of cells, including cell wall, chloroplast, cytoplasm, endoplasmic reticulum (ER), mitochondrion, nucleus, peroxisome and plasma membrane. In B73, 129 and 45 DEPs were predicted to locate in the chloroplast and cytoplasm, accounting for 49.62% and 17.31%, respectively (Figure 5a). In X178, 147 and 45 DEPs were predicted to locate in the chloroplast and cytoplasm, accounting for 46.52% and 14.24%, respectively (Figure 5b). Detailed subcellular distribution of the DEPs in B73 and X178 are illustrated in Figure 5c,d, respectively. Similar patterns of subcellular locations of most DEPs were observed in B73 and X178 infected by RBSDV, but the DEPs locating in ribosome were all up-regulated in B73, while all down-regulated in X178 following RBSDV infection (Figure 5c,d).

### 3.5. RBSDV Infection Regulates A Cell Number Regulator (CNR) Protein in B73, but Not in X178

Based on the above results, we speculated that enations on the abaxial surface of leaves might result from an increase in number of epidermal cells and phloem cells (Figure 2). Previously, several CNR genes have been identified in maize. CNR1 has been reported to negatively regulate plant and organ size through regulating cell number [35]. According to this proteomic analysis, a member of CNR protein family, CNR10 (GI: 226508610), was *c.* 3.2-fold down-regulated in leaves of RBSDV-infected B73 plants compared with those of mock-inoculated B73 plants, while the expression of CNR10 in leaves of RBSDV-infected X178 was not significantly different from that of the mock-inoculated X178 control plants (Table 1).

### 3.6. RBSDV Infection Represses Photosynthesis in Maize

The decrease of photosynthesis is commonly observed in virus-infected plants [36]. Consistently, our proteomic analysis revealed that four C4-dicarboxylic acid cycle-related and three Calvin-Benson cycle-related proteins were significantly down-regulated in B73 following RBSDV infection (Table 1; Figure 6a). These DEPs were phosphoenolpyruvate carboxylase (PEPC, GI: 257670508), pyruvate orthophosphate di-kinase 1 (PPDK1, GI: 219885341), malate dehydrogenase (NADP, GI: 414870017), NADP-dependent malic enzyme (NADP-ME, GI: 414875928), ribulose-1, 5-bisphosphate carboxylase/oxygenase (RuBisCo) large subunit (GI: 902230), fructose-1, 6-bisphospatase (FBP, 414879547) and transketolase 1 (TKT1, GI: 227483055) (Table 1). Though X178 shows resistance to RBSDV, the C4-dicarboxylic acid cycle-related and Calvin-Benson cycle-related DEPs, including PEPC, PPDK1, NADP, NADP-ME, RuBisCo large subunit, FBP, TKT1 and fructose-bisphosphate aldolase (FBA, GI: 413954603) were also down-regulated in RBSDV-infected X178 (Table 1; Figure 6a). Nevertheless, PEPC4 (GI: 414884997) was up-regulated in RBSDV-infected X178 but not identified in RBSDV-infected B73 (Table 1; Figure 6a). These results suggested that RBSDV infection might suppress photosynthesis in both susceptible line B73 and resistant line X178.

Leaf color changing from green to dark green is one of the typical symptoms on RBSDV-susceptible maize following RBSDV infection. However, in these dark green leaves, the content of chlorophyll a and chlorophyll b decreased significantly [37]. According to our proteomic analysis, five chlorophyll biosynthesis-related DEPs, glutamyl-tRNA ligase (GluRS, GI: 414864743), uroporphyrinogen decarboxylase (UROD, GI: 414866702), magnesium chelatase H subunit (ChlH, GI: 413919944), magnesium-protoporphyrin IX monomethyl ester oxidative cyclase (Mg-ProtoMe cyclase, GI: 413946872), NADPH-protochlorophyllide oxidoreductase (POR, GI: 414584771) and Chlorophyllase-1 (CLH1, GI: 414870069), were identified in B73, of which GluRS, UROD, ChlH and Mg-ProtoMe cyclase, which are upstream of the chlorophyll biosynthetic pathway, were significantly down-regulated, POR and CLH1, which are downstream of the chlorophyll biosynthetic pathway, were up-regulated (Table 1; Figure 6b). In contrast to B73, all the chlorophyll biosynthesis-related DEPs identified in X178, including glutamate-1-semialdehyde 2,1-aminomutase (GSA-AM, GI: 41394584), UROD, protoporphyrinogen oxidase (Protox, GI: 6715441), ChlH, Mg-ProtoMe cyclase, POR and CLH1, were up-regulated following RBSDV infection (Table 1; Figure 6b).

### 3.7. Altered Defense and Stress-Related Proteins

Plants are sessile organisms and suffer stresses from a range of biotic and abiotic factors. Therefore, they have evolved sophisticated defense strategies to respond to various biotic stresses and to adapt to an ever-changing environment. According to the above proteomic analysis, RBSDV infection led to a total of 43 and 66 stress-related proteins differentially expressed in B73 and X178, respectively (Figure 3d; Appendix A). In these DEPs, 26 DEPs were identified in both B73 and X178. A variety of antioxidant enzymes, such as lipoxygenases (LOX, GI: 8515851), peroxidase 12 (GI: 414878601) and peroxidase 45 (GI:413934711), were identified in RBSDV-infected B73 and X178. There are some antioxidant enzymes identified only in RBSDV-infected B73, including LOX (GI: 84626285), catalase 2 (GI: 414864621), catalase 3 isoform 2 (GI: 413926854), peroxidase 39 isoform 2 (GI: 413934538) and peroxidase 42 (GI: 221272351).

In this study, pathogenesis-related protein 10 (PR10, GI: 63079027) was identified to show opposite expression trends in RBSDV-infected B73 and X178. PR10 was *c.* 20-fold up-regulated in RBSDV-infected B73 while *c.* 2.78-fold down-regulated in RBSDV-infected X178 (Table 1). Meanwhile, pathogenesis-related protein 6 (PR6, GI: 77862323) was only identified in RBSDV-infected B73 with *c.* 15-fold up-regulation (Table 1).

Plant hormones play a crucial role in protecting plants against stresses. Two jasmonic acid (JA)-induced proteins (GI: 414589204, 413942196) were identified in RBSDV-infected B73 and X178, respectively. One JA-induced protein (GI: 414589204) was *c.* 3.47-fold up-regulated in RBSDV-infected B73, while it was not identified in RBSDV-infected X178. Meanwhile, the other JA-induced protein (GI: 413942196) was *c.* 8.63-fold up-regulated in RBSDV-infected X178, while it was not identified in RBSDV-infected B73. Cytokinin inducible protease 1 (GI: 413916758) was *c.* 3.25-fold down-regulated in RBSDV-infected X178, while it was not identified in RBSDV-infected B73 (Table 1).

Molecular chaperones regulate plant defense response by mediating signal transduction. We found that calreticulin2 (GI: 414884142), was *c.* 11.27- and *c.* 3.13-fold up-regulated in RBSDV-infected B73 and X178, respectively. Protein disulfide isomerase (PDI, GI: 59861261) was *c.* 4.02-fold up-regulated in RBSDV-infected B73. Heat shock proteins (HSPs), including Hsp70 protein (GI: 414866114) and putative heat shock protein 90 family protein (GI: 413925247), were *c.* 2.83- and *c.* 2.91-fold down-regulated in RBSDV-infected X178, respectively. Heat shock cognate 70 kDa protein 2 (GI: 414866114) was *c.* 1.84-fold up-regulated in RBSDV-infected B73 and *c.* 2.83-fold down-regulated in RBSDV-infected X178 (Table 1).

Many defense and stress-related proteins, such as auxin-binding protein (ABP, GI: 264278), plastid-lipid-associated protein 2 (GI: 414586941), filamentation temperature-sensitive H 2B isoform 2 (GI: 413943404), GDSL-motif protein lipase/hydrolase-like protein (GI: 413916727), ferredoxin-dependent glutamate synthase 1 isoform 2 (GI: 414887961) and putative FKBP-type peptidyl-prolyl cis-trans isomerase family proteins (GI: 414883868 and 413955031) were also identified in RBSDV-infected B73 or X178 (Table 1).

## 4. Discussion

RBSDV is the main causal agent of MRDD in China. Following infection, RBSDV causes typical enations on leaves of susceptible maize inbred line B73 plants but not on resistant inbred line X178 plants. By observation of the surface morphological structure and cross sections of enations, we found an excessive proliferation of epidermal cells on the abaxial surface of leaves infected by RBSDV. More importantly, a regulator of cell numbers CNR10 was significantly downregulated in RBSDV-infected B73, as seen through comparative proteomic analyses. Altogether, we speculated that RBSDV infection can regulate the expression of genes involved in cell proliferation, by which enation symptoms might be caused.

In this study, we chose leaf samples at the early stage of enation formation for comparative proteomic analyses of resistant maize inbred line X178 and susceptible line B73 plants. To minimize the variations caused by leaf development, the maturing zone [23] of maize leaves was harvested for iTRAQ assays. Moreover, protein fold change ratio >1.5 or <0.67 and false discovery rate (FDR)-corrected *p*-value < 0.05 were considered to be significantly up- or down-regulated upon RBSDV infection. Thus, these analyses could provide a solid basis for identification of the potential protein changes for enation formation during RBSDV infection.

According to this proteomic analysis, the expression of CNR10 was found to be significantly downregulated in RBSDV-infected B73, but not in RBSDV-infected X178. Maize *CNR* genes were identified through homology with a tomato (*Solanum lycopersicum*) *fw2.2*. Tomato *fw2.2* governs a quantitative trait locus that accounts for 30% of fruit size variation, with increased fruit size chiefly due to increased carpel ovary cell number [35,38,39]. In maize, 13 *CNRs* have been identified, of which encoded Cys-rich proteins contain the PLAC8 or DUF614 conserve motif [35]. *CNR1* negatively regulates plant and organ size by changing cell number but not cell size [35]. CNR13 is a maize MID-COMPLEMENTING ACTIVITY homolog, whose deletion mutant has an overall reduction in size and organ patterning defects. These severe phenotypes derive from defects in cell division, expansion and differentiation [40]. *CNR 10* and *CNR 3*, *9*, *12* make up a gene cluster all located on the same BAC AC186166 on chromosome 5. Though the function of *CNR10* is not yet clear, we presumed that it might be involved in regulating cell number in maize leaves and resulting in the formation of enation.

Virus infection generally represses photosynthesis in plants [36]. Similarly, photosynthesis-related proteins, such as PEPC, PPDK1, NADP, NADP-ME, RuBisCo large subunit, FBP, were down-regulated in RBSDV-infected B73 and X178 (Figure 6, Table 1, Appendix A). It has been reported that darkening of leaves in RBSDV-infected rice plants may be due to the abnormal expression of chloroplast- and photosynthesis-related proteins [37]. We found that chloroplast- and photosynthesis-related proteins were also down-regulated in RBSDV-infected X178 but showing normal leaves color. Based on these results, we speculated that RBSDV infection repressed photosynthesis in both susceptible plants and resistant plants.

It is reported that the phloem-limited RBSDV induces the formation of neoplastic phloem tissues to promote its multiplication in maize [34]. In this study, we found that viral RNAs in B73 leaves were 10^4^ times higher than those in X178 leaves. We speculate that the ability of RBSDV to induce the formation of enations in B73 might contribute to the high viral concentration.

In this study, 105 DEPs were identified in both B73 and X178 following RBSDV infection, of which 82 DEPs showed the same differential expression trend. These common DEPs with the same differential expression trend might be involved in maize defense responses to RBSDV infection, such as LOX (GI: 8515851), peroxidase 12 (GI: 414878601), calreticulin2, plastid-lipid-associated protein 2, filamentation temperature-sensitive H 2B isoform 2 and GDSL-motif protein lipase/hydrolase-like protein. Calreticulin (CRT), being a highly conserved and abundant multifunctional protein, is often associated with abiotic/biotic stress responses in plants [41,42,43]. *Arabidopsis* CRT2 is a negative regulator of plant innate immunity [44]. Maize CRT2 was up-regulated in both RBSDV-infected B73 and X178 with *c.* 11.27- and 3.13-fold change, respectively, suggesting an important role in maize response to RBSDV infection. The other 23 DEPs showed opposite differential expression trends in RBSDV-infected B73 and X178. These include proteins related to chlorophyll biosynthesis, such as UROD, ChlH and Mg-ProtoMe (Table 1 and Appendix A), which might be the reason why B73 and X178 plants showed different leaf colors following RBSDV infection. DEPs involved in plant defense response such as PR10, peroxidase 45, UDP-sulfo-quinovose synthase and heat shock cognate 70 kDa protein 2 (Table 1 and Appendix A), may contribute to the resistance of X178 or the susceptibility of B73. The DEPs only identified in RBSDV-infected B73 or X178 also play important roles in the resistance of X178 or the susceptibility of B73, such as PR6, ABP, PDI.

PR6 and PR10 participate in response to viral infection, which were induced strongly during the hypersensitive response (HR) against tobacco mosaic virus (TMV) isolate P_0_ in pepper (*Capsicum annuum* L. cv. Bugang) [45,46]. PR10 was reported to play a positive role in salt stress tolerance in *Arabidopsis* [47,48]. However, according to this proteomic analysis, PR10 was significantly up-regulated in susceptible B73, while down-regulated in resistant X178 following RBSDV infection. The mechanism of PR10 responding to virus infection should be further explored. PR6 protein was also induced in B73, while not in X178, in response to RBSDV infection. Previously, it has been proved that PR6 possesses proteinase inhibitor activity and was predicted to play a role in plant defense [49].

Some proteins associated with susceptibility or resistance to virus infection have been found through comparative proteomic analyses. In this work, ABP was significantly up-regulated in RBSDV-infected X178, but was not identified in RBSDV-infected B73. *ABP* gene underlies *Scmv2*, a major sugarcane mosaic virus (SCMV) resistance locus and the variation of *ABP* promoter affects maize SCMV resistance [50]. Further studies are needed to determine whether ABP plays a role in resistance to RBSDV in maize. PDI is known to be a principal catalyst for disulfide-linked protein folding in the ER lumen and is also a direct donor of disulfide bonds to nascent polypeptides through the thiol–disulfide exchange reaction [51]. It has been reported to be involved in *N*-gene-mediated immunity against TMV [52] and SCMV accumulation in maize [26]. PDI was up-regulated in RBSDV-infected B73 with *c.* 4.02-fold, but saw no change in RBSDV-infected X178. Moreover, PDI was also up-regulated in RBSDV-infected rice or SCMV-infected maize [26,53]. The up-regulated expression of PDI might be related to the large aggregation of RBSDV in infected cells.

In summary, we found that the excessive proliferation of epidermal cells and phloem cells is tightly associated with the formation of enations in RBSDV-infected susceptible line B73. Comparative proteomic analyses between resistant and susceptible maize plants were conducted at the early stage of enation forming. The proteins involved in metabolic process, response to stress and biosynthetic process showed significant changes in B73 and X178, suggesting that RBSDV infection disturbs these physiological processes in both the resistant and susceptible maize lines. Almost all the proteins involved in photosynthesis were significantly down-regulated after RBSDV infection in both resistant and susceptible lines, suggesting repressed photosynthesis in RBSDV-infected maize plants. In susceptible B73, the expression of CNR10 was significantly down-regulated, which might be involved in responsibility for the formation of enation.

## Figures and Tables

**Figure 1 viruses-14-02604-f001:**
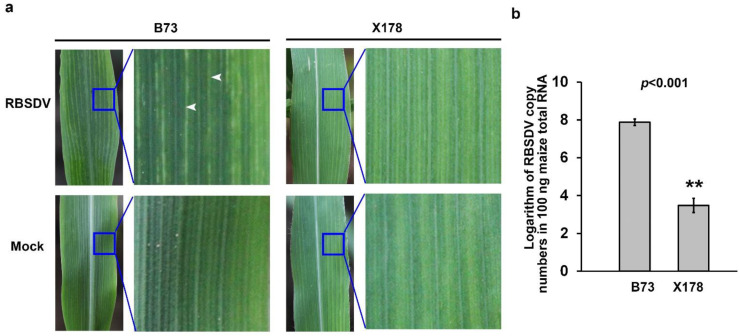
Infections on maize lines B73 and X178 leaves by rice black-streaked dwarf virus (RBSDV). (**a**) Enations formation on RBSDV-infected B73 leaves, but not on Mock or RBSDV-infected X178 leaves at 15 days post inoculation (dpi). White arrows indicate the enations in leaf blade abaxially along veins. (**b**) Reverse transcription-coupled quantitative polymerase chain reaction (RT-qPCR) results showed that RBSDV accumulation in the susceptible maize line B73 was significantly higher than that in the resistant maize line X178. Three independent experiments were conducted with three biological replicates per treatment in each experiment. Error bars represented the means ± SE. Significance of differences was determined using Student’s *t*-test. The asterisks indicate a statistically significant difference.

**Figure 2 viruses-14-02604-f002:**
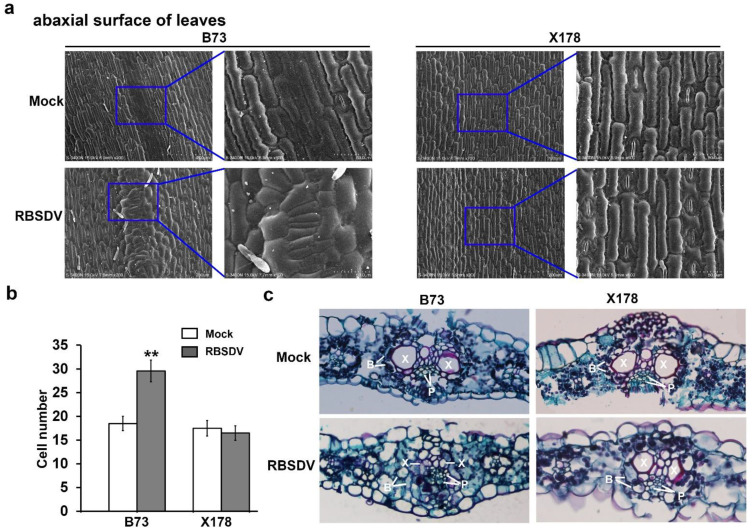
Microscopic structures of enations caused by RBSDV infection. (**a**) Scanning electron micrograph of abaxial surface of leaves on RBSDV-infected B73 or X178 plants and mock-inoculated B73 or X178 plants. (**b**) Statistic analysis of cell numbers of abaxial surface of leaves on RBSDV-infected B73 or X178 plants and mock-inoculated B73 or X178 plants. Three samples per treatment were used to count the number of cells. The asterisks indicate a statistically significant difference. (**c**) Micrograph of paraffin section of cross sections of leaves on RBSDV-infected B73 or X178 plants and mock-inoculated B73 or X178 plants. “B” indicates bundle sheath; “P” indicates phloem; “X” indicates xylem.

**Figure 3 viruses-14-02604-f003:**
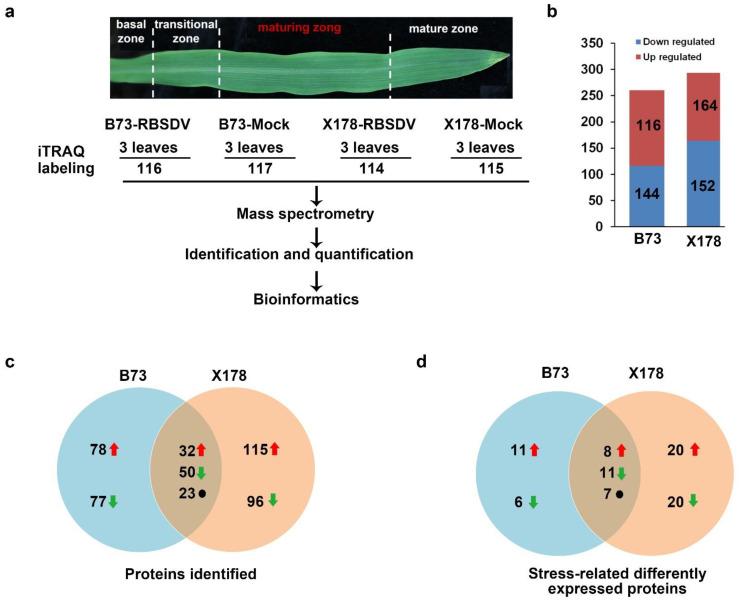
Flowchart for iTRAQ analyses and summary of differentially expressed proteins (DEPs) responsive to RBSDV infection in maturing zone of leaves on B73 and X178 plants. (**a**) Maturing zone of leaves in maize seedlings according to a previous report [23] was collected for iTRAQ. (**b**) Total number of up- and down-regulated proteins in leaves of RBSDV-infected B73 and X178, respectively. (**c**) Venn diagram representing the number of specific and common DEPs in RBSDV-infected B73 and X178. (**d**) Venn diagram representing the number of stress-related specific and common DEPs identified in RBSDV-infected B73 and X178. Red arrows represent up-regulated DEPs. Green arrows represent down-regulated DEPs. Black dots represent DEPs showing opposite expression trends in B73 and X178.

**Figure 4 viruses-14-02604-f004:**
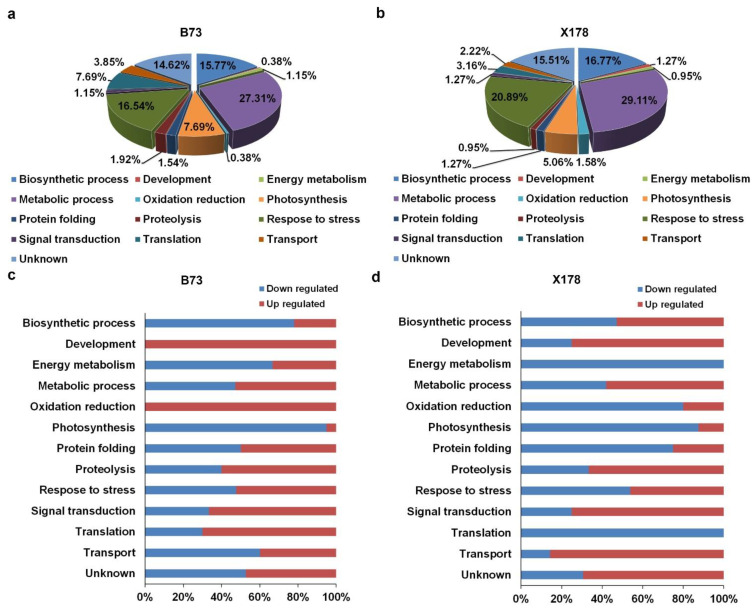
Functional categorizations of DEPs in leaves of maize B73 and X178 plants following RBSDV infection. (**a**,**b**) Distribution of DEPs in leaves of RBSDV-infected B73 or X178 in different functional categories. (**c**,**d**) Percentages of up- and down-regulated proteins in leaves of RBSDV-infected B73 or X178 in different functional categories.

**Figure 5 viruses-14-02604-f005:**
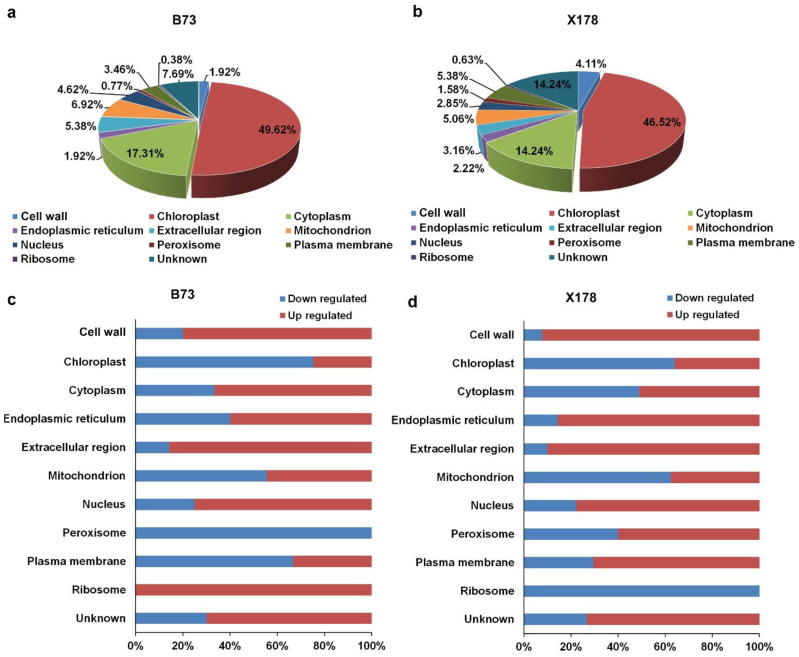
Subcellular localization patterns of DEPs in leaves of maize B73 and X178 following RBSDV infection. (**a**,**b**) Percentages of DEPs in leaves of RBSDV-infected B73 or X178 in different cell compartments. (**c**,**d**) Percentages of up- or down-regulated proteins in leaves of RBSDV-infected B73 or X178 in different cell compartments.

**Figure 6 viruses-14-02604-f006:**
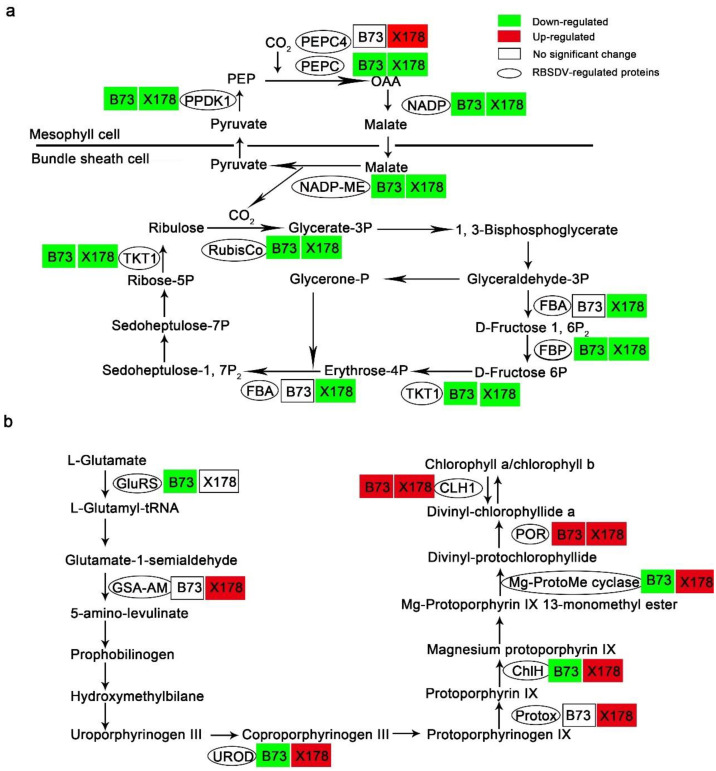
RBSDV infection altered proteins in photosynthesis and biosynthesis of chlorophyll a and b in leaves of maize lines B73 and X178. (**a**) RBSDV infection impeded the Calvin–Benson cycle and the C4-dicarboxylic acid cycle in both B73 and X178. PEPC, phosphoenolpyruvate carboxylase; PPDK1, pyruvate orthophosphate dikinase 1; NADP, malate dehydrogenase; NADP-ME, NADP-dependent malic enzyme; RuBisCo, ribulose-1, 5-bisphosphate carboxylase/oxygenase large subunit; FBA, fructose-bisphosphate aldolase; FBP, fructose-1, 6-bisphospatase; TKT1, transketolase 1. (**b**) RBSDV infection promoted chlorophyll a and b biosynthesis in X178 but not in B73. GluRS, glutamyl-tRNA ligase; GSA-AM, glutamate-1-semialdehyde 2,1-aminomutase; UROD, uroporphyrinogen decarboxylase; Protox, protoporphyrinogen oxidase; ChlH, magnesium chelatase H subunit; Mg-ProtoMe cyclase, magnesium-protoporphyrin IX monomethyl ester oxidative cyclase; POR, NADPH-protochlorophyllide oxidoreductase; CLH1, chlorophyllase-1.

**Table 1 viruses-14-02604-t001:** List of rice black-streaked dwarf virus (RBSDV) infection-regulated proteins in leaves of maize inbred lines B73 and X178 and their functions.

Accession	Name	Biological Process	Subcellular Location	Fold Change B73_RBSDV_:B73_Mock_	Fold Change X178_RBSDV_:X178_Mock_
Response to stress					
gi|8515851	* lipoxygenase	response to stress	cytoplasm	19.77	6.61
gi|84626285	lipoxygenase (LOX)	response to stress	unknown	2.99	-
gi|414864621	TPA: catalase2	response to stress	unknown	0.47	-
gi|413926854	catalase 3 isoform 2	response to stress	cell wall	2.40	-
gi|414878601	* TPA: peroxidase 12	response to stress	cell wall	3.53	2.05
gi|413934711	* peroxidase 45	response to stress	ER	3.34	0.27
gi|413934538	peroxidase 39 isoform 2	response to stress	ER	2.44	-
gi|221272351	peroxidase 42, Membrane-bound class III peroxidases3-1	response to stress	cell wall	3.60	-
gi|63079027	* pathogenesis-related protein 10	response to stress	unknown	20.70	0.36
gi|77862323	pathogenesis-related protein 6	response to stress	cell wall	15.00	-
gi|414884142	* TPA: calreticulin2	response to stress	ER	11.27	3.13
gi|59861261	protein disulfide isomerase	metabolic process	ER	4.02	-
gi|414866114	TPA: hsp70 protein	protein folding	cell wall	-	0.35
gi|413925247	putative heat shock protein 90 family protein	response to stress	cell wall	-	0.34
gi|414866114	* heat shock cognate 70 kDa protein 2	response to stress	cytoplasm	1.84	0.35
gi|48093330	chitinase	metabolic process	ER	19.41	-
gi|414586941	* TPA: plastid-lipid-associated protein 2	response to stress	chloroplast	0.41	0.30
gi|413943404	* filamentation temperature-sensitive H 2B isoform 2	response to stress	chloroplast	0.48	0.42
gi|413948935	* UDP-sulfo-quinovose synthase	response to stress	chloroplast	0.33	2.25
gi|413942196	jasmonate-induced protein	unknown	unknown	-	8.63
gi|414589204	TPA: jasmonate-induced protein	unknown	unknown	3.47	-
gi|413916758	cytokinin inducible protease1	response to stress	cell wall	-	0.31
gi|413916727	* GDSL-motif protein lipase/hydrolase-like protein	metabolic process	ER	4.88	25.59
gi|414887961	TPA: ferredoxin-dependent glutamate synthase1 isoform 2	response to stress	mitochondrion	-	0.38
gi|414883868	TPA: putative FKBP-type peptidyl-prolyl cis-trans isomerase family protein	response to stress	chloroplast	-	0.22
gi|264278	auxin-binding protein	signal transduction	ER	-	2.68
Symptom-related					
gi|226508610	cell number regulator 10	unknown	unknown	0.31	-
Photosynthesis related					
gi|257670508	* phosphoenolpyruvate carboxylase (PEPC)	biosynthetic process	cytoplasm	0.42	0.42
gi|414884997	phosphoenolpyruvate carboxylase (PEPC4)	metabolic process	cytoplasm	-	2.70
gi|219885341	* pyruvate orthophosphate di-kinase 1 (PPDK 1)	photosynthesis	cytoplasm	0.30	0.40
gi|414870017	* malate dehydrogenase (NADP)	metabolic process	cytoplasm	0.38	0.33
gi|414875928	* NADP-dependent malic enzyme (NADP-ME)			0.44	0.29
gi|902230	* ribulose-1,5-bisphosphate carboxylase/oxygenase large subunit (RuBisCo)	photosynthesis	chloroplast	0.35	0.38
gi|414879547	* fructose-1,6-bisphospatase (FBP)	metabolic process	cytoplasm	0.25	0.29
gi|227483055	* transketolase 1 (TKT1)	metabolic process	chloroplast	0.22	0.57
gi|763035	* glyceraldehyde-3-phosphate dehydrogenase	metabolic process	chloroplast	0.43	0.45
gi|413944536	* 6-phosphofructo-2-kinase/fructose-2, 6-bisphosphatase	metabolic process	cytoplasm	0.47	0.46
gi|413954603	fructose-bisphosphate aldolase (FBA)	metabolic process	cytoplasm	-	0.44
gi|414864743	glutamyl-tRNA ligase cytoplasmic (GluRS)	metabolic process	cytoplasm	0.40	-
gi|414866702	* uroporphyrinogen decarboxylase (UROD)	biosynthetic process	chloroplast	0.56	1.85
gi|413919944	* magnesium-chelatase H subunit (ChlH)	biosynthetic process	chloroplast	0.21	3.08
gi|413946872	* magnesium-protoporphyrin IX monomethyl ester oxidative cyclase (Mg-ProtoMe cyclase)	biosynthetic process	chloroplast	0.46	1.98
gi|414584771	* NADPH-protochlorophyllide oxidoreductase (POR)	photosynthesis	chloroplast	1.51	4.74
gi|414870069	* chlorophyllase-1 (CLH1)	metabolic process	unknown	3.37	4.13
gi|6715441	protoporphyrinogen oxidase (Protox)	biosynthetic process	chloroplast	-	1.75
gi|41394584	glutamate-1-semialdehyde 2,1-aminomutase (GSA-AM)	biosynthetic process	chloroplast	-	1.77

Asterisks represent the proteins identified in both B73 and X178 plants following RBSDV infection.

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
