# Peer review of "Comparative Proteomic Analyses of Susceptible and Resistant Maize Inbred Lines at the Stage of Enations Forming following Infection by Rice Black-Streaked Dwarf Virus"

_viruses, 2022, doi:10.3390/v14122604_

Round 1

Reviewer 1 Report

The authors Wang and colleagues performed comparative proteomic analyses study with B73 (susceptible to RBSDV) and X178 plants (resistant to RBSDV), and identified some proteins involved in enation formation against RBSDV infection by isobaric tags for relative and absolute quantification (iTRAQ) technique. This is very interesting but I think the author must improve something.

 Major comments,

The authors mentioned 23 DEPs only showed opposite up-or down-regulated in B73 and X178 in lines 231-232, but I do not see it in other section. The reviewer think those genes are very important for further study. It is these oppositely expressed genes that may lead to the development of these differential symptoms. The authors should add some information about this.

Additionally, draw a schematic representation of putative host proteins involved in enation formation against RBSDV infection may better clear to see.

Finally, I ask the authors to revise the whole manuscript to weaken statements which have not been proven yet. So far, it is only a speculation, but still outstanding and interesting work.

Minor comments

In lines 23-25, I really fail to see how CNR10 could be explain the formation of enations. The authors do not show any data that such experiments have been performed. I would delete this statement (or at least provide an qPCR for it).

In result 3.1, the authors only checked accumulation of viral RNA in this section, but the accumulation of viral proteins is an important indicator of viral infection, western blot may need to assay.

In figure2, how many samples were analyzed for cell number? Please clarify.

What’s the mean of “c.27.31%” in line 248? And also other similar number in the text. Please clarify.

Reviewer 2 Report

RBSDV is the main pathogen causing maize rough dwarf disease, which is also capable infection of the rice and wheat. Its causes big economic losses in crop production, and is most valuable research object. In this paper, wang et al. compared the difference in protein expression between the RBSDV-susceptible (B73) and RBSDV-resistance maize inbred lines (X178) under RBSDV-infection and during the enations formation using the iTRAQ proteomic analysis. The manuscript also outlined the differentially expressed proteins in metabolic process, stress responses, biosynthetic process, and determined the possible factor, cell number regulator 10 (CNR10), which involved in the enations formation by increasing of epidermal cell and phloem cell number. The present study provides the reference for study from the phenotype to underlying molecular mechanism of plant virus-host interactions. The tittle of the manuscript is “Comparative proteomic analyses of susceptible and resistant 2 maize inbred lines at the stage of enations forming following 3 infections by rice black-streaked dwarf virus”, which lead us thinking that the author will exploring the underlying molecular mechanism of the enations formation under RBSDV infection. However, the detailed experiment validation of series genes involved in the enations forming were not performed, the only related CNR10 was also outlined in the manuscript. I strongly suggest the author further explored the growth and cell number related genes, and using the qRT-PCR to validate the results. Overall, I think this manuscript is suitable for publication but only after some revisions.

Major revisions:

1.       Add the content of gene expression validation using the qRT-PCR;

2.       Explore more growth and cell number related genes using the GO and KEGG analyses, and identified the changed pathways in host plant under RBSDV infection. And this difference not between RBSDV-susceptible (B73) and RBSDV-resistance maize inbred lines (X178), it should be the difference between mock and RBSDV-infected maize inbred line B73.

3.       The language should be polished by professional agent or native English speaker.

Minor revisions

1.       The reference should be formatted, and refer the detailed demands of the journal of viruses.

Round 2

Reviewer 1 Report

I appreciate that my comments and suggestions were taken into consideration. In my opinion this R2 version can be accepted for publication.